# Learning Taxonomic Trees with Hierarchical Representation Regularization for Large Multimodal Models

**Hulingxiao He**[1]   **Zhi Tan**[1]   **Yuxin Peng**[1]

## Abstract

Taxonomies provide key information about the semantic relationships between concepts and the inherent organization of vision and language. Despite their impressive capabilities, large multimodal models (LMMs) often lack taxonomic knowledge, leading to low hierarchical visual recognition (HVR) consistency. These models typically only rely on language modeling objectives during fine-tuning and lack explicit taxonomy-aware regularization. To address this, we propose *Hierarchical Representation Regularization (HiR²)*, a simple plug-and-play regularizer that improves hierarchical consistency in LMMs. Specifically, we introduce a semantic-aware visual tree construction framework that extracts coarse-to-fine visual features from intermediate LLM layers guided by textual cues. The regularizer combines two complementary objectives: a taxonomic entailment loss that enforces hierarchy via hyperbolic entailment cones in the Lorentz model, and a discriminative dispersive loss that promotes angular separation of semantically similar embeddings on the unit sphere *without disturbing the radial hierarchical structure*. Extensive experiments demonstrate that HiR² effectively captures taxonomic structures across diverse LMMs and fine-tuning methods. Code is available at https://github.com/PKU-ICST-MIPL/HiR2_ICML2026.

## 1. Introduction

The real world is not binary, but governed by taxonomies. Taxonomies encode rich semantic relations between concepts and reflect the inherent organization of vision and language. For example, hierarchical visual recognition (HVR)

[1]Wangxuan Institute of Computer Technology, Peking University. Correspondence to: Yuxin Peng <pengyuxin@pku.edu.cn>.

*Proceedings of the 43rd International Conference on Machine Learning*, Seoul, South Korea. PMLR 306, 2026. Copyright 2026 by the author(s).

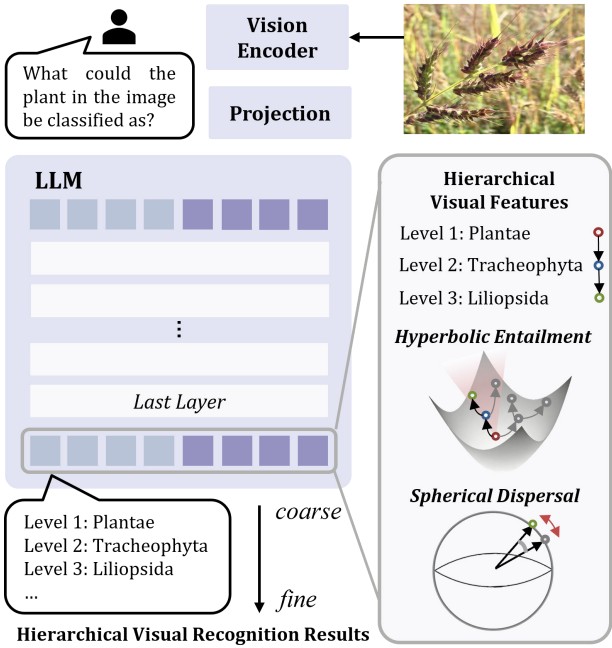

*Figure 1.* Hierarchical Representation Regularization encourages the intermediate representations to learn tree-like structures according to taxonomy beyond language modeling loss, thereby enhancing hierarchical visual recognition capability.

aims at categorizing real-world concepts that are naturally organized at multiple levels of abstraction. These relations form tree-structured hierarchies, where general concepts such as "plants" reside near the root, while more specific categories such as "trees" or "flowers" appear at deeper levels. Effectively modeling such structures is fundamental to hierarchical understanding of the visual world.

While existing large multimodal models (LMMs) (Bai et al., 2025; He et al., 2025a; Ma et al., 2026) have demonstrated strong performance in fine-grained visual recognition (FGVR) (Peng et al., 2025), they remain weak in HVR and often fail to maintain hierarchical consistency (Tan et al., 2026). This limitation largely stems from their training objectives: LMMs are primarily optimized with a language modeling loss for next-token prediction, without any explicit *taxonomy-aware regularization* imposed on the visual representations used for language generation. This paradigm con-

trasts sharply with prior work on discriminative vision (Park et al., 2025c) or vision-language models (Stevens et al., 2024; Pal et al., 2024; Wei et al., 2025), where hierarchical representation learning has been a central research topic for enhancing the prediction consistency.

Motivated by hierarchical representation learning, we are *the first* to explore its integration into LMMs and propose *Hierarchical Representation Regularization* (HiR$^2$), a flexible and plug-and-play framework that injects taxonomic knowledge into LMMs without introducing additional learnable parameters (Xiao et al., 2026). As illustrated in Figure 1, the key idea is to complement the standard language modeling objective with auxiliary losses that explicitly regularize internal visual representations to follow taxonomic tree structures. This formulation introduces two main challenges: (1) extracting hierarchical visual features with coarse-to-fine semantics from LMMs, and (2) designing learning objectives that faithfully encode taxonomic relations.

To address the first challenge, we develop a semantic-aware framework that leverages hidden states from the last layer of the LLM to construct hierarchical visual features. Specifically, a non-parametric cross-attention module is introduced, where textual features at each hierarchy level act as queries and visual token embeddings serve as keys and values, producing hierarchy-specific visual representations. These representations naturally form a semantic-aware visual tree with coarse-to-fine semantics.

To address the second challenge, we introduce two complementary learning objectives to structure the taxonomic tree. The first is a *taxonomic entailment loss*, which enforces geometric containment constraints in the hyperbolic Lorentz model, ensuring that child concepts (e.g., *Flower*) lie within the entailment cone of their parents (e.g., *Plant*). Hyperbolic space is particularly suitable for this purpose due to its exponential volume growth. The second is a *discriminative dispersive loss*, which increases angular separation between visually ambiguous sibling categories while preserving the radial hierarchy, thereby improving fine-grained discriminability without disrupting the taxonomic structure. The training procedure follows standard LMM fine-tuning practices, only adding additional regularization terms that incur little computational overhead. With its minimalist and self-contained design, HiR$^2$ demonstrates that injecting taxonomy knowledge via hierarchical representation learning can substantially benefit LMMs.

We evaluate the effectiveness and generality of HiR$^2$ through extensive experiments. Results show consistent improvements over strong LMM baselines such as Qwen2.5-VL (Bai et al., 2025), Qwen2-VL (Wang et al., 2024a), Intern3.5-VL-1B (Wang et al., 2025), and LLaVA-OV-1.5-4B(An et al., 2025), across both supervised fine-tuning (SFT) and dynamic fine-tuning (DFT) (Wu et al.,

2025) methods on various taxonomies like iNaturalist-2021 (iNat21) (Van Horn et al., 2021) and CUB-200-2011 (CUB-200) (Wah et al., 2011). These findings highlight the simplicity, effectiveness, and broad applicability of HiR$^2$ for learning taxonomic structures in LMMs. Our contributions are summarized as follows:

(1) This work presents the first principled study that introduces hierarchical representation learning as a regularization mechanism for large multimodal models beyond standard language modeling objectives.

(2) A semantic-aware visual tree construction framework is proposed to extract hierarchical visual representations with coarse-to-fine semantics from intermediate layers of large multimodal models.

(3) Two complementary objectives are developed: a taxonomic entailment loss that encodes hierarchical priors in hyperbolic space, and a dispersive loss that improves fine-grained discrimination via angular separation while preserving the radial hierarchical structure.

## 2. Related Work

**Hierarchical Visual Recognition.** HVR (Silla & Freitas, 2011; Kosmopoulos et al., 2015) plays a central role in understanding both visual (Yi et al., 2022; Park et al., 2025b; Zeng et al., 2024; Sinha et al., 2024; Chen et al., 2022; Park et al., 2025a) and language concepts (Zhou et al., 2020; Wang et al., 2022; Zhou et al., 2025; He et al., 2024). Recent studies reveal that CLIP-style models (Radford et al., 2021a) often lack taxonomic consistency (Wu et al., 2024; Geng et al., 2023). To address this, (Wu et al., 2024) evaluate CLIP across multiple granularity levels and propose hierarchy-consistent prompt tuning, while others enhance CLIP via hyperbolic embeddings (Pal et al., 2024) or graph-based learning (Xia et al., 2023). (Novack et al., 2023) further show that hierarchical supervision can improve zero-shot classification. Beyond CLIP, (Zhang et al., 2024) first identify the limitations of LMMs in FGVR, later extended by (Liu et al., 2024). (He et al., 2025b) attribute this issue to insufficient exposure to class names during pretraining. Subsequent evaluations consider both closed-set (Yu et al., 2025a; Geigle et al., 2024; He et al., 2026a) and open-world settings (Conti et al., 2025; He et al., 2026a), with (Snæbjarnarson et al., 2025) advocating taxonomic similarity over exact label matching. Most recently, (Tan et al., 2026) explicitly analyze the hierarchical understanding of LMMs, highlighting persistent challenges in hierarchical consistency and leaf-level accuracy. (He et al., 2026b) propose representation alignment to inject taxonomic knowledge from discriminative models into generative LMMs.

**Learning on Hyperbolic Manifolds.** Hyperbolic manifolds provide an effective geometry for modeling hierar-

chical structures, motivating a growing body of work on hyperbolic neural networks that incorporate non-Euclidean operations for representation learning (Guo et al., 2022; Shimizu et al., 2020; He et al., 2025d; Malik et al., 2025; Skopek et al., 2020; Gao et al., 2021; Yu et al., 2025b; Fan et al., 2025). They have been successfully applied across graphs (Fu et al., 2023; 2024; Malik et al., 2025), text (He et al., 2025c), images (Wang et al., 2024c; Franco et al., 2024; Li et al., 2025b; Gao et al., 2023; Li et al., 2025a) and videos (Long et al., 2020; Hong et al., 2023). Recent advances further extend hyperbolic learning to multimodal settings by combining entailment-aware objectives with CLIP-style training (Ramasinghe et al., 2024; Desai et al., 2023; Pal et al., 2025; Wang et al., 2024b), and by scaling hyperbolic representations to large vision–language models (Mandica et al., 2024; Wei et al., 2025).

**Representation Learning as Auxiliary Tasks.** Beyond standard pre-training and fine-tuning, representation learning is often incorporated as an auxiliary objective jointly optimized with the main task (Ye et al., 2022). Supervised contrastive learning (Khosla et al., 2020) augments classification with contrastive objectives, while SLIP (Mu et al., 2022) extends CLIP (Radford et al., 2021b) by adding a parallel self-supervised signal. In image generation, REPA (Yu et al., 2024) aligns intermediate generative features with those of a frozen encoder, later improved by SARA (Chen et al., 2025) through structural and adversarial alignment, and extended to multimodal settings by SoftREPA (Lee et al., 2025). For LMMs, VIRAL (Yoon et al., 2025) aligns internal visual representations with pretrained vision encoders to inject complementary visual knowledge, while JARVIS (Caffagni et al., 2025) employs a masked predictive objective to align predicted intermediate representations with those of a target encoder using a single context block.

# 3. Problem Setting and Preliminaries

In this section, we introduce the problem setting and the preliminaries considered in this paper.

**Problem Setting.** Conventional visual recognition typically assumes a flat label space, where each image $x \in \mathcal{X}$ is assigned a single label $y \in \mathcal{Y}$. However, real-world visual concepts are often structured hierarchically, with labels organized in a taxonomy $\mathcal{T} = (\mathcal{Y}, \mathcal{E})$ (Park et al., 2025a; Wu et al., 2024; Yi et al., 2022; Xia et al., 2023), such as a tree or a directed acyclic graph (DAG). Each directed edge $(y_i, y_j) \in \mathcal{E}$ encodes a parent–child relationship, where $y_i$ is the parent of $y_j$. In hierarchical visual recognition (HVR), the goal is to predict not only a leaf label $y \in \mathcal{Y}_{\text{leaf}}$, but also its full ancestral path $(y_0, y_1, \ldots, y_L)$ from root to leaf.

**Hyperbolic Manifold.** Unlike Euclidean spaces with zero curvature, a hyperbolic manifold is a smooth Riemannian manifold with constant negative curvature $-\kappa$ ($\kappa > 0$) (Lee, 2006). Following prior work (Cannon et al., 1997), we adopt the *Lorentz model* for hyperbolic geometry due to its computational efficiency and numerical stability. Formally, the $d$-dimensional Lorentz model is defined as

$$\mathbb{L}^{d,\kappa} = \left\{ \boldsymbol{p} \in \mathbb{R}^{d+1} \,\middle|\, \langle \boldsymbol{p}, \boldsymbol{p} \rangle_{\mathbb{L}} = -\frac{1}{\kappa}, \; p_0 > 0 \right\}, \quad (1)$$

which corresponds to the upper sheet of a two-sheeted hyperboloid embedded in $(d+1)$-dimensional Minkowski space.

Each point $\boldsymbol{p} \in \mathbb{L}^{d,\kappa}$ is represented as $\boldsymbol{p} = [p_0, \tilde{\boldsymbol{p}}]$, where $p_0 \in \mathbb{R}$ is the *time-like* component and $\tilde{\boldsymbol{p}} \in \mathbb{R}^d$ is the *space-like* component. The Lorentzian inner product $\langle \cdot, \cdot \rangle_{\mathbb{L}}$ between two points $\boldsymbol{p} = [p_0, \tilde{\boldsymbol{p}}]$ and $\boldsymbol{q} = [q_0, \tilde{\boldsymbol{q}}]$ is defined as

$$\langle \boldsymbol{p}, \boldsymbol{q} \rangle_{\mathbb{L}} = -p_0 q_0 + \langle \tilde{\boldsymbol{p}}, \tilde{\boldsymbol{q}} \rangle_{\mathbb{E}}, \quad (2)$$

where $\langle \cdot, \cdot \rangle_{\mathbb{E}}$ denotes the Euclidean inner product. The induced Lorentzian norm is $\|\boldsymbol{p}\|_{\mathbb{L}} = \sqrt{|\langle \boldsymbol{p}, \boldsymbol{p} \rangle_{\mathbb{L}}|}$.

**Distance.** The geodesic (shortest-path) distance between two points $\boldsymbol{p}, \boldsymbol{q} \in \mathbb{L}^{d,\kappa}$ is given by

$$d_{\mathbb{L}}(\boldsymbol{p}, \boldsymbol{q}) = \frac{1}{\sqrt{\kappa}} \operatorname{arccosh}(-\kappa \langle \boldsymbol{p}, \boldsymbol{q} \rangle_{\mathbb{L}}). \quad (3)$$

**Tangent Space.** Each point $\boldsymbol{p} \in \mathbb{L}^{d,\kappa}$ is associated with a $d$-dimensional tangent space $T_{\boldsymbol{p}} \mathbb{L}^{d,\kappa}$, which is a Euclidean vector space providing a local linear approximation. Any ambient vector $\boldsymbol{u} \in \mathbb{R}^{d+1}$ can be projected onto $T_{\boldsymbol{p}} \mathbb{L}^{d,\kappa}$ via

$$\operatorname{proj}_{\boldsymbol{p}}^{\kappa}(\boldsymbol{u}) = \boldsymbol{u} + \kappa \boldsymbol{p} \langle \boldsymbol{p}, \boldsymbol{u} \rangle_{\mathbb{L}}. \quad (4)$$

**Exponential Map.** The exponential map $\exp_{\boldsymbol{p}}^{\kappa} : T_{\boldsymbol{p}} \mathbb{L}^{d,\kappa} \to \mathbb{L}^{d,\kappa}$ projects a tangent vector $\boldsymbol{v} \in T_{\boldsymbol{p}} \mathbb{L}^{d,\kappa}$ onto the manifold:

$$\exp_{\boldsymbol{p}}^{\kappa}(\boldsymbol{v}) = \cosh\left(\sqrt{\kappa} \|\boldsymbol{v}\|_{\mathbb{L}}\right) \boldsymbol{p} + \frac{\sinh(\sqrt{\kappa} \|\boldsymbol{v}\|_{\mathbb{L}})}{\sqrt{\kappa} \|\boldsymbol{v}\|_{\mathbb{L}}} \boldsymbol{v}. \quad (5)$$

In particular, the exponential map at the origin $\boldsymbol{0} = (\sqrt{1/\kappa}, 0, \ldots, 0)^{\top}$ is used to embed Euclidean features into hyperbolic space.

**Logarithmic Map.** Conversely, the logarithmic map $\log_{\boldsymbol{q}}^{\kappa} : \mathbb{L}^{d,\kappa} \to T_{\boldsymbol{q}} \mathbb{L}^{d,\kappa}$ maps a point $\boldsymbol{p} \in \mathbb{L}^{d,\kappa}$ to the tangent space at $\boldsymbol{q}$:

$$\log_{\boldsymbol{q}}^{\kappa}(\boldsymbol{p}) = \frac{\operatorname{arccosh}(-\kappa \langle \boldsymbol{q}, \boldsymbol{p} \rangle_{\mathbb{L}})}{\sqrt{(\kappa \langle \boldsymbol{q}, \boldsymbol{p} \rangle_{\mathbb{L}})^2 - 1}} \operatorname{proj}_{\boldsymbol{q}}^{\kappa}(\boldsymbol{p}). \quad (6)$$

**Hyperbolic Entailment Cones.** Hyperbolic entailment cones associate each point $\boldsymbol{p} = [p_0, \tilde{\boldsymbol{p}}] \in \mathbb{L}^{d,\kappa}$ with a cone-shaped region $\omega(\boldsymbol{p})$, such that any $\boldsymbol{c} = [c_0, \tilde{\boldsymbol{c}}] \in \omega(\boldsymbol{p})$ is interpreted as a child of $\boldsymbol{p}$. The cone's half-aperture is

$$\omega(\boldsymbol{p}) = \arcsin\left(\frac{2\gamma}{\sqrt{\kappa} \|\tilde{\boldsymbol{p}}\|}\right), \quad (7)$$

where $\gamma = 0.1$ in our experiments.

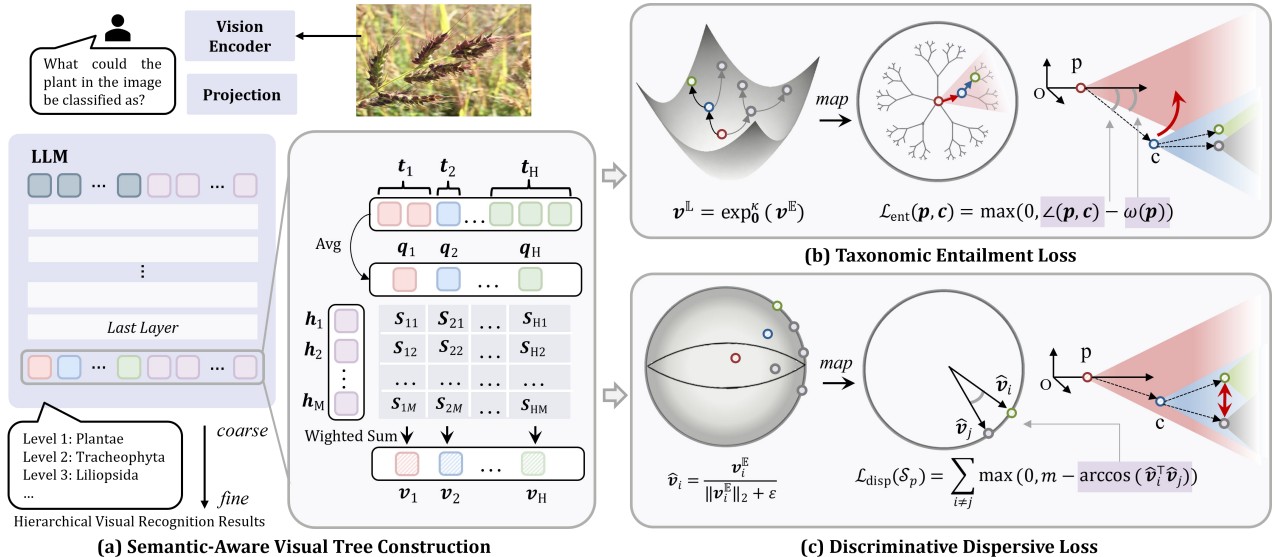

*Figure 2.* Overview of the HiR$^2$ framework. (a) Constructing hierarchical visual features from the last LLM layer. (b) Optimizing with a taxonomic entailment loss to enforce the hierarchy. (c) Optimizing with a discriminative dispersive loss for distinguishing semantically-similar categories.

## 4. Methodology

Motivated by the potential of representation learning for improving hierarchical consistency in LMMs, we propose *Hierarchical Representation Regularization* (HiR$^2$), a plug-and-play framework that explicitly injects taxonomic structure into the intermediate visual representations of LMMs. As illustrated in Figure 2, HiR$^2$ constructs hierarchy-aware visual features from internal hidden states, and enforces both hierarchical entailment and sibling-level discriminability through geometry-aware regularization objectives. HiR$^2$ can be easily computed without introducing additional learning parameters, as shown in Algorithm 1.

### 4.1. Semantic-aware Visual Tree Construction

Motivated by the cross-modal information flow mechanism of LMMs (Zhang et al., 2025), we leverage hidden states from intermediate (or last) layers to construct hierarchical visual features corresponding to different levels of the taxonomic tree.

**Hierarchical Visual Feature Extraction.** We denote the textual embedding sequence corresponding to different levels of categories as $[\boldsymbol{T}_1; \boldsymbol{T}_2; \ldots; \boldsymbol{T}_H]$, where $H$ denotes the depth of the textual hierarchy. For the $i$-th level,

$$\boldsymbol{T}_i = [\boldsymbol{t}_i(1); \boldsymbol{t}_i(2); \ldots; \boldsymbol{t}_i(N_i)], \tag{8}$$

and $N_i$ is the token length of the $i$-th level category name.

Let $\boldsymbol{H} = [\boldsymbol{h}_1; \boldsymbol{h}_2; \ldots; \boldsymbol{h}_M] \in \mathbb{R}^{M \times D}$ denote the hidden states corresponding to image tokens at the same transformer layer, where $M$ is the number of image tokens and $D$ is the

embedding dimension. We first obtain a level-wise textual query embedding by average pooling over tokens belonging to the same hierarchy level:

$$\boldsymbol{q}_i = \frac{1}{N_i} \sum_{j=1}^{N_i} \boldsymbol{t}_i(j), \quad i = 1, \ldots, H. \tag{9}$$

To extract visual features aligned with different hierarchy levels, we design a nonparametric cross-attention mechanism where each textual query $\boldsymbol{q}_i$ attends to image token features. Specifically, the attention weights over image tokens are computed as

$$\mathcal{S}_{ij} = \frac{\exp\left(\boldsymbol{q}_i^\top \boldsymbol{h}_j / \sqrt{D}\right)}{\sum_{k \in \mathcal{I}} \exp\left(\boldsymbol{q}_i^\top \boldsymbol{h}_k / \sqrt{D}\right)}, \tag{10}$$

where $\mathcal{I}$ denotes the set of positions corresponding to image tokens, ensuring that attention is restricted to the visual modality.

The hierarchical visual embedding for the $i$-th level is then obtained as a weighted aggregation of image token features:

$$\boldsymbol{v}_i = \sum_{j \in \mathcal{I}} \mathcal{S}_{ij} \boldsymbol{h}_j, \quad i = 1, \ldots, H. \tag{11}$$

As a result, we obtain a hierarchy-aware visual feature set $\{\boldsymbol{v}_i^{\mathbb{E}}\}_{i=1}^H$, where each $\boldsymbol{v}_i^{\mathbb{E}} \in \mathbb{R}^D$ captures hierarchy-aligned visual semantics in Euclidean space, serving as a geometry-agnostic intermediate representation before hierarchical regularization.

**Curvature-aware Scaling.** Before mapping Euclidean visual features into hyperbolic space, we apply a *curvature-aware scaling* operation to normalize embedding magnitudes at the batch level. Given Euclidean visual embeddings $\boldsymbol{v}_i^{\mathbb{E}} \in \mathbb{R}^D$, we first compute their $\ell_2$ norms and estimate the batch-wise average norm

$$\bar{r} = \frac{1}{|\mathcal{B}|} \sum_{i \in \mathcal{B}} \|\boldsymbol{v}_i^{\mathbb{E}}\|_2, \tag{12}$$

where $\mathcal{B}$ denotes the set of embeddings within the current batch. We then rescale all embeddings using a shared scaling factor

$$s = \frac{\rho}{\sqrt{\kappa}\,(\bar{r} + \varepsilon)}, \tag{13}$$

where $\kappa$ is the target hyperbolic curvature, $\rho \in (0, 1)$ is a predefined target ratio, and $\varepsilon$ is a small constant for numerical stability. The scaled embeddings are given by

$$\tilde{\boldsymbol{v}}_i^{\mathbb{E}} = s \cdot \boldsymbol{v}_i^{\mathbb{E}}. \tag{14}$$

Importantly, the average norm $\bar{r}$ is detached from the computational graph, so the scaling factor does not introduce additional gradients. This batch-level normalization aligns the expected embedding radius with the curvature-dependent scale $1/\sqrt{\kappa}$, stabilizing subsequent exponential mapping into the Lorentz model and preventing excessive norm growth during training.

**Exponential Map.** After Curvature-aware Scaling, we map all Euclidean features into the hyperbolic embeddings of Lorentz model. This transition is achieved via the exponential map (Nickel & Kiela, 2018) at the origin $\boldsymbol{0}$, denoted as $\exp_{\boldsymbol{0}}^{\kappa} : T_{\boldsymbol{0}}\mathbb{L}^{d,\kappa} \to \mathbb{L}^{d,\kappa}$. For ease of expression, $\boldsymbol{v}^{\mathbb{E}}$ is used to denote any Euclidean visual features, *e.g.*, $[\boldsymbol{v}_1^{\mathbb{E}}; \boldsymbol{v}_2^{\mathbb{E}}; \ldots; \boldsymbol{v}_H^{\mathbb{E}}]$. Its corresponding hyperbolic embedding $\boldsymbol{v}^{\mathbb{L}} \in \mathbb{L}^{d,\kappa}$ is computed as $\boldsymbol{v}^{\mathbb{L}} = \exp_{\boldsymbol{0}}^{\kappa}(\boldsymbol{v}^{\mathbb{E}})$.

### 4.2. Taxonomic Entailment Loss

To explicitly encode the hierarchical structure of taxonomic trees, we devise a *taxonomic entailment loss* $\mathcal{L}_{\text{ent}}$ defined in hyperbolic space. Unlike Euclidean or spherical geometries whose volume growth is polynomial or constant with respect to radius, hyperbolic space exhibits exponential volume expansion, closely matching the branching property of tree-structured data. This geometric property allows hierarchical depth to be naturally encoded by radial distance, making hyperbolic space particularly suitable for modeling parent-child relations in deep taxonomies.

Formally, this loss leverages hyperbolic entailment cones (Ganea et al., 2018; Desai et al., 2023), wherein a parent concept $p$ geometrically entails its child concept $q$ by constraining them to lie within its cone-shaped region. We apply this loss to preserve both semantic hierarchy and conceptual

**Algorithm 1** LMM Fine-tuning + HiR$^2$

**Require:** Outputs `outputs`, labels `labels`, image hidden states $\boldsymbol{H}$, text hierarchy $\{\boldsymbol{T}_i\}$, taxonomy $\mathcal{T}$
**Ensure:** Total loss $\mathcal{L}_{\text{total}}$
1: Compute language modeling loss $\mathcal{L}_{\text{LM}}$
2: **for** each selected image–text layer **do**
3:    Construct hierarchical Euclidean visual features $\{\boldsymbol{v}_i^{\mathbb{E}}\}_{i=1}^H$ via cross-attention
4:    Apply curvature-aware scaling to $\boldsymbol{v}_i^{\mathbb{E}}$
5:    Map to hyperbolic space $\boldsymbol{v}_i^{\mathbb{L}} = \exp_{\boldsymbol{0}}^{\kappa}(\boldsymbol{v}_i^{\mathbb{E}})$
6:    Accumulate taxonomic entailment loss $\mathcal{L}_{\text{ent}}$ over $(\boldsymbol{v}_i^{\mathbb{L}}, \boldsymbol{v}_{i+1}^{\mathbb{L}})$
7:    Project $\boldsymbol{v}_i^{\mathbb{E}}$ to angular components $\hat{\boldsymbol{v}}_i$
8:    Accumulate parent-aware dispersive loss $\mathcal{L}_{\text{disp}}$ with memory banks
9: **end for**
10: Apply scheduled activation $\alpha$ to $\mathcal{L}_{\text{disp}}$
11: $\mathcal{L}_{\text{total}} = \mathcal{L}_{\text{LM}} + \lambda_{\text{ent}}\mathcal{L}_{\text{ent}} + \lambda_{\text{disp}}\mathcal{L}_{\text{disp}}$

hierarchy. For each parent–child pair $(\boldsymbol{p}, \boldsymbol{c})$, we penalize child embeddings outside the cone:

$$\mathcal{L}_{\text{ent}}(\boldsymbol{p}, \boldsymbol{c}) = \max\big(0,\ \angle(\boldsymbol{p}, \boldsymbol{c}) - \omega(\boldsymbol{p})\big), \tag{15}$$

where

$$\angle(\boldsymbol{p}, \boldsymbol{c}) = \arccos\left(\frac{p_0 + \kappa\langle\boldsymbol{p}, \boldsymbol{c}\rangle_{\mathbb{L}} c_0}{\|\tilde{\boldsymbol{c}}\|\sqrt{(\kappa\langle\boldsymbol{p}, \boldsymbol{c}\rangle_{\mathbb{L}})^2 - 1}}\right). \tag{16}$$

The total entailment loss across hierarchy levels is

$$\mathcal{L}_{\text{ent}} = \sum_{i=1}^{H-1} \mathcal{L}_{\text{ent}}\left(\boldsymbol{v}_i^{\mathbb{L}}, \boldsymbol{v}_{i+1}^{\mathbb{L}}\right). \tag{17}$$

### 4.3. Discriminative Dispersive Loss

While the hyperbolic taxonomic entailment loss effectively preserves hierarchical depth, it does not explicitly enforce discrimination among sibling categories under the same parent. In deep hierarchies, such sibling embeddings may collapse along similar directions, leading to ambiguous fine-grained representations.

To address this issue without interfering with hyperbolic radial hierarchy, we introduce a *discriminative dispersive loss* defined on a spherical space, where embeddings are constrained to lie on a unit hypersphere and discrimination is governed purely by angular separation. The fixed radius and bounded volume of spherical space make it particularly suitable for modeling sibling-level diversity while remaining complementary to hyperbolic hierarchy modeling.

**Spherical Angular Separation.** By projecting Euclidean visual embeddings onto a unit hypersphere, we isolate their

Table 1. Variants of the dispersive loss.

| Variant | $\delta(\cdot, \cdot)$ | Space / Reference |
|---|---|---|
| I | $d_{\mathbb{H}}(z_i, z_j)$ | $\mathbb{H}$ |
| II | $d_{\text{ang}}(\log_{\mathbf{0}}(z_i), \log_{\mathbf{0}}(z_j))$ | $T_{\mathbf{0}}\mathbb{H}$ |
| III | $d_{\text{ang}}(\log_p(z_i), \log_p(z_j))$ | $T_p\mathbb{H}$ |
| IV | $d_{\text{ang}}(z_i - p, z_j - p)$ | $\mathbb{R}^D$ |
| Ours | $d_{\text{ang}}(\hat{v}_i, \hat{v}_j), \ \|\hat{v}\|_2 = 1$ | $\mathbb{S}^{D-1}$ |

angular components by $\hat{v}_i = \frac{v_i^{\mathbb{E}}}{\|v_i^{\mathbb{E}}\|_2 + \varepsilon}$. This spherical normalization removes radial degrees of freedom, ensuring that the dispersive objective depends solely on angular relationships and does not introduce implicit hierarchical ordering.

**Sibling-only Repulsion.** On the resulting unit hypersphere, we define sibling-level repulsion purely in terms of angular distance. Formally, for parent $p$, let $\mathcal{S}_p = \{\hat{v}_i \mid \text{parent}(v_i) = p\}$. We encourage angular separation among siblings with a margin-based objective:

$$\mathcal{L}_{\text{disp}}(\mathcal{S}_p) = \sum_{i \neq j} \max\bigl(0, m - \arccos(\hat{v}_i^\top \hat{v}_j)\bigr), \quad (18)$$

while ignoring embeddings from different parents.

**Parent-aware Memory Bank.** To enhance stability and long-term diversity, each parent $p$ maintains a memory bank $\mathcal{M}_p$ of past embeddings. After angular projection, interactions with memory extend the loss:

$$\mathcal{L}_{\text{disp}}(\mathcal{S}_p, \mathcal{M}_p) = \sum_{\hat{v} \in \mathcal{S}_p} \sum_{\hat{u} \in \mathcal{M}_p} \max\bigl(0, m - \arccos(\hat{v}^\top \hat{u})\bigr). \ (19)$$

**Scheduled Activation.** The dispersive loss is gradually activated with a linearly increasing factor $\alpha \in [0, 1]$:

$$\mathcal{L}_{\text{disp}} = \alpha \sum_p \mathcal{L}_{\text{disp}}(\mathcal{S}_p, \mathcal{M}_p), \quad (20)$$

aggregating over all parents in the taxonomy.

**Total Loss.** Finally, the overall objective combines language modeling and the two hierarchical regularizers:

$$\mathcal{L}_{\text{total}} = \mathcal{L}_{\text{LM}} + \lambda_{\text{ent}}\mathcal{L}_{\text{ent}} + \lambda_{\text{disp}}\mathcal{L}_{\text{disp}}. \quad (21)$$

### 4.4. Variants of Dispersive Loss

We investigate several alternative designs of the dispersive loss by varying the geometric space in which sibling separation is measured, while keeping the same margin-based formulation. As summarized in Table 5, these variants include: (1) Variant I: hyperbolic geodesic distance $d_{\mathbb{L}}(z_i, z_j)$ in the Lorentz model. (2) Variant II: angular distance in the tangent space at the hyperbolic origin where $d_{\text{ang}}(\mathbf{u}, \mathbf{v}) = \arccos \frac{\mathbf{u}^\top \mathbf{v}}{\|\mathbf{u}\|_2 \|\mathbf{v}\|_2}, \quad \mathbf{u}, \mathbf{v} \in \mathbb{R}^D$. (3) Variant

III: parent-centered tangent-space angular separation, and (4) Variant IV: parent-relative spherical angular separation. While these designs are geometrically motivated, they either entangle angular dispersion with radial hierarchy, rely on global reference points, or introduce distortion through repeated logarithmic mappings.

To analyze the interaction between dispersive objectives and the radial-angular decomposition of hyperbolic embeddings, we unify the input space for all sibling embeddings as the unit hypersphere

$$\hat{v} \in \mathbb{S}^{D-1} = \left\{ u \in \mathbb{R}^D \ \middle| \ \|u\|_2 = 1 \right\}, \quad (22)$$

where $\hat{v}$ is obtained via angular normalization of Euclidean visual features, and let $z \in \mathbb{H}^D$ denote the corresponding hyperbolic embeddings under the exponential map $\exp_0^\kappa$ with radial component $\rho(z)$.

**Theorem 4.1.** *For dispersive objectives defined using hyperbolic geodesic distances or tangent-space angular measures, optimization generally induces gradients along the radial directions of hyperbolic embeddings, even when the inputs are spherical embeddings $\hat{v}_i, \hat{v}_j \in \mathbb{S}^{D-1}$.*

Formally, let $\hat{v}_i, \hat{v}_j \in \mathbb{S}^{D-1}$ be sibling embeddings, and $z_i, z_j \in \mathbb{H}^D$ their hyperbolic counterparts. If the loss depends on $d_{\mathbb{H}}(z_i, z_j)$ or on angular measures computed via $\log_c(\cdot)$ for any reference point $c$, then, except for degenerate configurations, $\nabla_{\rho(z_i)}\mathcal{L} \neq 0$ and $\nabla_{\rho(z_j)}\mathcal{L} \neq 0$.

**Theorem 4.2.** *The proposed spherical angular dispersive loss, defined on $\hat{v} \in \mathbb{S}^{D-1}$, does not induce gradients on the radial components of hyperbolic embeddings.*

Specifically, for any sibling embedding $\hat{v}_i \in \mathbb{S}^{D-1}$ with $z_i = \exp_0^\kappa(\hat{v}_i)$,

$$\frac{\partial \mathcal{L}_{\text{disp}}}{\partial \rho(z_i)} = 0, \quad (23)$$

ensuring that the hierarchical structure encoded by hyperbolic radii remains unchanged.

**Theorem 4.3.** *Increasing spherical angular separation between sibling embeddings on $\mathbb{S}^{D-1}$ increases their angular separation in hyperbolic space.*

For $\hat{v}_i, \hat{v}_j \in \mathbb{S}^{D-1}$ and $z_i = \exp_0^\kappa(\hat{v}_i), z_j = \exp_0^\kappa(\hat{v}_j)$, enlarging $\angle(\hat{v}_i, \hat{v}_j)$ monotonically increases $\angle(z_i, z_j)$, improving sibling discriminability without affecting radial ordering.

**Intuition.** Hyperbolic hierarchy is primarily encoded by radial depth $\rho(z)$, while sibling discrimination is governed by angular structure on $\mathbb{S}^{D-1}$. Most existing dispersive variants entangle these two factors through hyperbolic distances or tangent-space mappings. By operating on spherical embeddings, our loss preserves radial hierarchy while enhancing angular separation, yielding stable and hierarchically consistent representations in deep taxonomies.

*Table 2.* Performance evaluation on iNat-Plant and iNat-Animal with different hierarchical distributions. The best results are in green.

| Dataset | Hierarchy Distribution | Methods | Base | | | | Novel | | | |
|---|---|---|---|---|---|---|---|---|---|---|
| | | | HCA ↑ | POR ↑ | S-POR ↑ | TOR ↑ | HCA ↑ | POR ↑ | S-POR ↑ | TOR ↑ |
| iNat-Plant | 5-14-85-286 -1702-4271 | SFT | 9.98 | 63.63 | 46.01 | 45.69 | 10.86 | 63.74 | 46.15 | 45.16 |
| | | SFT+Ours | 11.90 ↑1.92 | 66.65 ↑3.01 | 50.87 ↑4.86 | 49.76 ↑4.07 | 13.06 ↑2.20 | 67.16 ↑3.42 | 50.69 ↑4.54 | 49.69 ↑4.53 |
| | | DFT | 8.15 | 62.45 | 40.47 | 41.98 | 9.32 | 63.26 | 41.35 | 42.64 |
| | | DFT+Ours | 11.66 ↑3.51 | 66.07 ↑3.62 | 49.83 ↑9.36 | 48.81 ↑6.83 | 12.87 ↑3.55 | 66.72 ↑3.46 | 50.12 ↑8.77 | 49.10 ↑6.46 |
| iNat-Animal | 6-27-152-715 -2988-5388 | SFT | 14.06 | 71.41 | 60.40 | 58.03 | 15.74 | 72.12 | 61.42 | 58.86 |
| | | SFT+Ours | 18.07 ↑4.01 | 73.35 ↑1.94 | 64.27 ↑3.87 | 61.17 ↑3.14 | 17.60 ↑1.86 | 73.77 ↑1.65 | 64.63 ↑3.21 | 61.49 ↑2.63 |
| | | DFT | 14.69 | 71.90 | 60.27 | 58.81 | 16.52 | 72.65 | 61.64 | 60.04 |
| | | DFT+Ours | 16.33 ↑1.64 | 72.54 ↑0.64 | 62.44 ↑2.17 | 59.58 ↑0.77 | 17.04 ↑0.52 | 73.16 ↑0.51 | 63.01 ↑1.37 | 60.49 ↑0.45 |

*Table 3.* Performance evaluation on iNat-Animal with more base models. The best results are in green.

| Model | Methods | Base | | | | Novel | | | |
|---|---|---|---|---|---|---|---|---|---|
| | | HCA ↑ | POR ↑ | S-POR ↑ | TOR ↑ | HCA ↑ | POR ↑ | S-POR ↑ | TOR ↑ |
| Intern3.5-VL-1B | SFT | 2.04 | 51.13 | 37.07 | 32.32 | 1.63 | 51.33 | 36.58 | 32.44 |
| | SFT+Ours | 3.04 ↑1.00 | 53.86 ↑2.73 | 40.64 ↑3.57 | 35.67 ↑3.35 | 2.93 ↑1.30 | 54.17 ↑2.84 | 40.80 ↑4.22 | 36.06 ↑3.62 |
| LLaVA-OV-1.5-4B | SFT | 5.75 | 61.68 | 51.55 | 47.06 | 6.05 | 61.50 | 51.62 | 47.29 |
| | SFT+Ours | 6.49 ↑0.74 | 63.45 ↑1.77 | 54.20 ↑2.65 | 49.70 ↑2.64 | 6.09 ↑0.04 | 63.22 ↑1.72 | 53.89 ↑2.27 | 49.66 ↑2.37 |
| Qwen2-VL-2B | SFT | 6.12 | 61.88 | 51.22 | 47.18 | 5.31 | 61.77 | 51.38 | 46.77 |
| | SFT+Ours | 6.42 ↑0.30 | 63.14 ↑1.26 | 54.15 ↑2.93 | 49.03 ↑1.85 | 5.94 ↑0.63 | 63.08 ↑1.31 | 54.32 ↑2.94 | 49.13 ↑2.36 |

# 5. Experiments

## 5.1. Experimental Setup

**Datasets and Tasks.** To rigorously evaluate the effectiveness of HiR$^2$, we conduct comprehensive experiments on the iNaturalist-2021 (iNat21) dataset (Van Horn et al., 2021). Following (Tan et al., 2026), the dataset is divided into two taxonomies, *Plant* and *Animal*, comprising 4,271 and 5,388 leaf nodes, respectively, organized into six hierarchical levels. Unless otherwise specified, each taxonomy is evenly split into base and novel classes, and all experiments are performed under the 1-shot training setting. Models are trained on base classes and evaluated on both base classes (base-to-base) and novel classes (base-to-novel).

**Evaluation Metrics.** We assess LMMs' prediction consistency in hierarchical visual recognition task using four metrics: Hierarchical Consistent Accuracy (HCA) (Wu et al., 2024; Park et al., 2025b), Point-Overlap Ratio (POR) (Yi et al., 2022), Strict Point-Overlap Ratio (S-POR) (Tan et al., 2026), and Top Overlap Ratio (TOR) (Wu et al., 2024). Detailed definitions are provided in Appendix B.1.

We defer additional evaluation settings and implementation details to Appendix B.2 and Appendix B.3, respectively.

## 5.2. Main Results

Table 2 compares HiR$^2$ with widely used LMM fine-tuning methods on the Qwen2.5-VL-3B (Bai et al., 2025) model across different taxonomies. Several key observations can be drawn. (1) Compatibility with different LMM fine-tuning methods: HiR$^2$ consistently improves performance under both supervised fine-tuning (SFT) and dynamic fine-tuning

(DFT) (Wu et al., 2025). The gains in all metrics are observed across all settings, indicating that the proposed hierarchical regularization is complementary to existing LMM fine-tuning strategies rather than being tied to a specific training paradigm. (2) Robustness across taxonomies and base-to-novel generalization: Consistent improvements are achieved on both iNat-Plant and iNat-Animal, despite their distinct taxonomic structures and visual characteristics. Moreover, HiR$^2$ improves performance on both base and novel classes in the base-to-novel evaluation, demonstrating strong generalization under distribution shift and robustness to unseen categories. (3) Balancing hierarchical consistency and discriminability: HCA, the most strict metric, requires predictions to be correct at all taxonomic levels simultaneously, and thus cannot be improved by enhancing hierarchical consistency or per-level discrimination alone. The consistent HCA gains indicate that HiR$^2$ effectively balances these two factors, yielding coherent and discriminative representations throughout the entire taxonomy.

## 5.3. Ablation Studies

To validate the contributions of individual components in HiR$^2$, we conduct extensive ablation studies. Unless otherwise stated, all ablation experiments are performed on Qwen2.5-VL-3B using the iNat-Animal taxonomy.

**Evaluation on Other Models.** Thus far, our experiments have focused on Qwen2.5-VL-3B model. In Table 3, we extend our evaluation to Intern3.5-VL-1B (Wang et al., 2025), LLaVA-OV-1.5-4B(An et al., 2025), and Qwen2-VL-2B(?). Once again, HiR$^2$ consistently improves performance over the baselines across all metrics.

**Evaluation on Other Taxonomies.** We further examine

*Table 4.* Performance evaluation on CUB-200 with four-level hierarchy. The best results are in green.

| Dataset | Model | Methods | Base | | | | Novel | | | |
|---|---|---|---|---|---|---|---|---|---|---|
| | | | HCA ↑ | POR ↑ | S-POR ↑ | TOR ↑ | HCA ↑ | POR ↑ | S-POR ↑ | TOR ↑ |
| CUB-200 | Qwen2.5-VL-3B | SFT | 9.36 | 53.01 | 15.81 | 33.08 | 8.76 | 51.48 | 16.28 | 33.54 |
| | | SFT+Ours | 35.38 ↑26.02 | 76.30 ↑23.29 | 63.38 ↑47.57 | 60.52 ↑27.44 | 32.83 ↑24.07 | 74.79 ↑23.31 | 59.06 ↑42.78 | 58.25 ↑24.71 |

*Table 5.* Ablation study on variants of dispersive loss.

| Variant | Base | | Novel | |
|---|---|---|---|---|
| | HCA | $Acc_{leaf}$ | HCA | $Acc_{leaf}$ |
| I | 15.33 | 39.78 | 16.90 | 40.62 |
| II | 16.73 | 39.85 | 16.86 | 40.59 |
| III | 17.92 | 40.67 | 18.23 | 42.00 |
| IV | 17.89 | 40.74 | 18.01 | 41.14 |
| Ours | 18.07 | 41.37 | 17.60 | 42.52 |

*Table 6.* Ablation study on key design components.

| $\mathcal{L}_{ent}$ | $\mathcal{L}_{disp}$ | Layer Index | Base | Novel |
|---|---|---|---|---|
| Zero-shot | | | 15.92 | 39.00 |
| SFT | | | 14.06 | 40.37 |
| *Ablation on different regularization loss* | | | | |
| ✓ | | 28 | 17.89 | 40.78 |
| ✓ | ✓ | 28 | 18.07 | 41.37 |
| *Ablation on different regularization layer* | | | | |
| ✓ | ✓ | 7 | 14.55 | 40.52 |
| ✓ | ✓ | 14 | 14.36 | 39.04 |
| ✓ | ✓ | 21 | 11.32 | 37.63 |
| ✓ | ✓ | 28 | 18.07 | 41.37 |
| ✓ | ✓ | 7,14,21,28 | 15.18 | 39.93 |
| ✓ | ✓ | all layers | 12.06 | 38.81 |

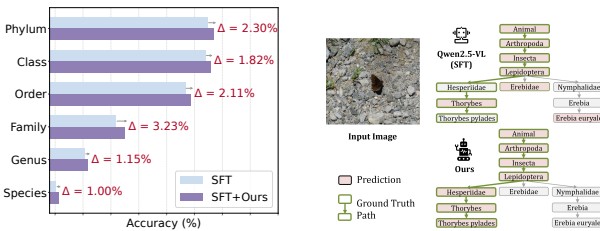

*Figure 3.* Left: Per-level accuracy on iNat-Animal base classes; Right: Qualitative comparsion.

insufficient to induce taxonomic structure. Introducing $\mathcal{L}_{ent}$ enforces hierarchical consistency and yields clear gains, while $\mathcal{L}_{disp}$ further improves fine-grained separability. Combining both losses achieves the best overall performance, demonstrating their complementary effects in HiR$^2$.

**Target Regularization Layer.** We further investigate the effect of applying regularization at different layers. As shown in Table 6, performance varies across layers, with regularizing the last layer producing the strongest results.

**Per-level Accuracy.** Figure 3 (Left) shows that HiR$^2$ consistently improves accuracy across all hierarchy levels, with larger gains at coarser levels. This indicates that the improvements mainly stem from enhanced hierarchical consistency rather than purely improved fine-grained discrimination, underscoring the benefit of our approach beyond FGVR.

**Qualitative Results.** As shown in Figure 3 (Right), HiR$^2$ produces consistent predictions along the taxonomy, whereas the SFT-based baseline yields inconsistent outputs, highlighting the importance of explicitly learning taxonomic structures for hierarchical visual recognition.

More ablation studies are in Appendix B.4.

## 6. Conclusion

In this work, we have proposed Hierarchical Representation Regularization (HiR$^2$), an approach that regularizes the internal representations of LMMs for learning taxonomic trees. Our experiments across various taxonomies, base models, and fine-tuning strategies demonstrate the effectiveness of HiR$^2$ in improving the performance of HVR. Importantly, a key principle guiding our design is to introduce minimal or no interference with the language modeling process of the original training objective. This allows HiR$^2$ compatible with any LMM and fine-tuning strategy like supervised fine-tuning, emphasizing its widespread applicability.

whether HiR$^2$ generalizes to different taxonomies. Experiments are conducted on the CUB-200-2011 (CUB-200) dataset (Wah et al., 2011), which adopts a four-level hierarchy where leaf nodes correspond to bird common names rather than scientific names (Tan et al., 2026). We use all training samples from 100 base classes and evaluate performance on both base and novel classes. As shown in Table 4, HiR$^2$ consistently improves all evaluation metrics, indicating its adaptability across datasets and its effectiveness as a regularizer beyond standard fine-tuning.

**Variants of Dispersive Loss.** Table 5 compares different dispersive loss variants that measure sibling separation in different geometric spaces. Variants defined in hyperbolic space or tangent spaces (I–IV) improve HCA to varying degrees, but yield limited gains on the leaf-node accuracy $Acc_{leaf}$ due to interference with the radial structure that encodes taxonomy. In contrast, our spherical angular dispersive loss achieves the best consistency–discriminability trade-off by computing angular separation directly on the unit hypersphere, demonstrating the strongest improvements on $Acc_{leaf}$, especially at the deepest level of the hierarchy.

**Effects of $\mathcal{L}_{ent}$ and $\mathcal{L}_{disp}$.** Table 6 shows that SFT on VQA data with taxonomic labels alone may even hurt the discrimination of similar base classes, suggesting that SFT is

## Acknowledgements

This work was supported by the grants from Beijing Natural Science Foundation (L247006) and the National Natural Science Foundation of China (62525201, 62132001, 62432001).

## Impact Statement

This paper presents work whose goal is to advance the field of Machine Learning. There are many potential societal consequences of our work, none which we feel must be specifically highlighted here.

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

# A. Theoretical Analysis of Dispersive Loss

In this section, we provide a rigorous theoretical analysis of different variants of the dispersive loss under hyperbolic geometry. We focus on how various dispersive objectives interact with the radial–angular decomposition of hyperbolic embeddings, and formally justify why the proposed spherical angular formulation uniquely preserves radial hierarchy while improving sibling-level discrimination.

**Notation.** Let $\hat{v} \in \mathbb{S}^{D-1}$ denote a unit-norm spherical embedding,

$$\mathbb{S}^{D-1} = \left\{ u \in \mathbb{R}^D \mid \|u\|_2 = 1 \right\}. \tag{24}$$

Its corresponding hyperbolic embedding is obtained via the exponential map at the origin,

$$z = \exp_0^{\kappa}(v) \in \mathbb{H}^D, \tag{25}$$

where $v \in \mathbb{R}^D$ is a Euclidean vector with direction $\hat{v} = v/\|v\|$. We denote by $\rho(z) = \|v\|$ the radial component of $z$, which encodes hierarchical depth in hyperbolic space.

Throughout this section, we consider sibling embeddings $\hat{v}_i, \hat{v}_j \in \mathbb{S}^{D-1}$ and their hyperbolic counterparts $z_i, z_j \in \mathbb{H}^D$.

**Theorem 4.1 (Radial Entanglement of Existing Dispersive Variants).** *For dispersive objectives defined using hyperbolic geodesic distances $d_{\mathbb{H}}(z_i, z_j)$ or angular measures computed via logarithmic maps $\log_c(\cdot)$ at any reference point $c$, optimization generally induces non-zero gradients on the radial components $\rho(z_i)$ and $\rho(z_j)$, even when the inputs are spherical embeddings $\hat{v}_i, \hat{v}_j \in \mathbb{S}^{D-1}$.*

**Proof.** We analyze the gradient of representative dispersive objectives with respect to the radial component $\rho(z_i)$.

*Case (i): Hyperbolic geodesic distance.* The hyperbolic distance between $z_i$ and $z_j$ satisfies

$$d_{\mathbb{H}}(z_i, z_j) = \frac{1}{\sqrt{\kappa}} \operatorname{arccosh}\left(\cosh(\sqrt{\kappa}\rho_i)\cosh(\sqrt{\kappa}\rho_j) - \sinh(\sqrt{\kappa}\rho_i)\sinh(\sqrt{\kappa}\rho_j)\langle \hat{v}_i, \hat{v}_j \rangle\right), \tag{26}$$

where $\rho_i = \rho(z_i)$.

Taking the partial derivative with respect to $\rho_i$ yields

$$\frac{\partial d_{\mathbb{H}}}{\partial \rho_i} = \frac{\sinh(\sqrt{\kappa}\rho_i)\cosh(\sqrt{\kappa}\rho_j) - \cosh(\sqrt{\kappa}\rho_i)\sinh(\sqrt{\kappa}\rho_j)\langle \hat{v}_i, \hat{v}_j \rangle}{\sqrt{\left(\cosh(\sqrt{\kappa}\rho_i)\cosh(\sqrt{\kappa}\rho_j) - \sinh(\sqrt{\kappa}\rho_i)\sinh(\sqrt{\kappa}\rho_j)\langle \hat{v}_i, \hat{v}_j \rangle\right)^2 - 1}}. \tag{27}$$

Except for degenerate configurations (e.g., identical directions and radii), this derivative is non-zero. Hence, any dispersive loss depending on $d_{\mathbb{H}}$ induces gradients along radial directions.

*Case (ii): Tangent-space angular objectives.* Consider an angular dispersive term defined as

$$\delta = \angle(\log_c(z_i), \log_c(z_j)), \tag{28}$$

where $c$ is either the hyperbolic origin or a parent node.

The logarithmic map admits the form

$$\log_c(z_i) = \alpha_i\left(z_i + \kappa\langle c, z_i\rangle c\right), \tag{29}$$

where the scalar coefficient

$$\alpha_i = \frac{\operatorname{arccosh}(-\kappa\langle c, z_i\rangle)}{\sqrt{(\kappa\langle c, z_i\rangle)^2 - 1}} \tag{30}$$

depends explicitly on the Lorentz inner product $\langle c, z_i\rangle$, which in turn depends on $\rho(z_i)$.

Consequently, both the direction and magnitude of $\log_c(z_i)$ vary with $\rho(z_i)$, and by the chain rule,

$$\frac{\partial \delta}{\partial \rho(z_i)} \neq 0 \tag{31}$$

in general.

Combining the above cases proves the claim. $\square$

**Theorem 4.2 (Radial Invariance of Spherical Angular Dispersive Loss).** *The proposed spherical angular dispersive loss, defined on $\hat{v} \in \mathbb{S}^{D-1}$, induces zero gradient on the radial components of the corresponding hyperbolic embeddings.*

**Proof.** The proposed dispersive loss is defined as

$$\mathcal{L}_{\text{disp}} = \mathcal{L}\big(\angle(\hat{\boldsymbol{v}}_i, \hat{\boldsymbol{v}}_j)\big), \qquad \hat{\boldsymbol{v}}_i, \hat{\boldsymbol{v}}_j \in \mathbb{S}^{D-1}. \tag{32}$$

By construction, $\hat{\boldsymbol{v}}_i$ depends only on the direction of $\boldsymbol{v}_i$ and is invariant to its norm. Therefore,

$$\frac{\partial \mathcal{L}_{\text{disp}}}{\partial \|\boldsymbol{v}_i\|} = 0. \tag{33}$$

Since the hyperbolic radius satisfies $\rho(\boldsymbol{z}_i) = \|\boldsymbol{v}_i\|$, the chain rule gives

$$\frac{\partial \mathcal{L}_{\text{disp}}}{\partial \rho(\boldsymbol{z}_i)} = \frac{\partial \mathcal{L}_{\text{disp}}}{\partial \|\boldsymbol{v}_i\|} \cdot \frac{\partial \|\boldsymbol{v}_i\|}{\partial \rho(\boldsymbol{z}_i)} = 0. \tag{34}$$

Hence, the proposed loss does not perturb hyperbolic radial hierarchy. $\square$

**Theorem 4.3 (Angular Consistency Between Sphere and Hyperbolic Space).** *Increasing spherical angular separation between sibling embeddings $\hat{\boldsymbol{v}}_i, \hat{\boldsymbol{v}}_j \in \mathbb{S}^{D-1}$ monotonically increases their angular separation in hyperbolic space.*

**Proof.** Let $\boldsymbol{z}_i = \exp_{\boldsymbol{0}}^{\kappa}(\boldsymbol{v}_i)$ and $\boldsymbol{z}_j = \exp_{\boldsymbol{0}}^{\kappa}(\boldsymbol{v}_j)$ with fixed radii $\rho_i$ and $\rho_j$. Their Lorentz inner product satisfies

$$-\kappa\langle \boldsymbol{z}_i, \boldsymbol{z}_j \rangle = \cosh(\sqrt{\kappa}\rho_i)\cosh(\sqrt{\kappa}\rho_j) - \sinh(\sqrt{\kappa}\rho_i)\sinh(\sqrt{\kappa}\rho_j)\langle \hat{\boldsymbol{v}}_i, \hat{\boldsymbol{v}}_j \rangle. \tag{35}$$

For fixed $\rho_i$ and $\rho_j$, the right-hand side is a strictly decreasing affine function of $\langle \hat{\boldsymbol{v}}_i, \hat{\boldsymbol{v}}_j \rangle$. Since

$$\angle(\hat{\boldsymbol{v}}_i, \hat{\boldsymbol{v}}_j) = \arccos\big(\langle \hat{\boldsymbol{v}}_i, \hat{\boldsymbol{v}}_j \rangle\big), \tag{36}$$

increasing spherical angular separation strictly increases the hyperbolic angular separation between $\boldsymbol{z}_i$ and $\boldsymbol{z}_j$.

Thus, optimizing spherical angular dispersion directly enhances sibling discriminability in hyperbolic space without altering radial ordering. $\square$

# B. More Experimental Details and Results

## B.1. Evaluation Metrics

To comprehensively evaluate model performance, we focus on the hierarchical consistency of predictions (Wu et al., 2024; Park et al., 2025b), complemented by the leaf-level classification accuracy (Zhang et al., 2024; Liu et al., 2024; He et al., 2025b), which serves as the upper bound of hierarchical consistency. The evaluation metrics are detailed below.

**Hierarchical Consistent Accuracy (HCA)** (Wu et al., 2024; Park et al., 2025b). This metric is defined as

$$\text{HCA} = \frac{1}{N} \sum_{i=1}^{N} \prod_{j=1}^{L^i} \mathbf{1}\left[ f_\theta\left(x^i; \mathcal{Y}_j\right) = y_j^i \right], \tag{37}$$

Here, $N$ is the number of test samples, $L^i$ is the depth of the hierarchy for the $i$-th input $x^i$, and $\mathcal{Y}j$ denotes the label set at level $j$. HCA computes the proportion of samples whose predicted paths exactly match the ground truth from root to leaf. It is therefore a stricter criterion than flat accuracy and serves as our primary evaluation metric for hierarchical classification.

**Point-Overlap Ratio (POR)** (Yi et al., 2022). It measures hierarchical performance beyond strict correctness, defined as:

$$\text{POR} = \frac{1}{N} \sum_{i=1}^{N} \frac{\sum_{j=1}^{L_i} \mathbf{1}\left[ f_\theta\left(x_i; \mathcal{Y}_j\right) = y_j^i \right]}{L_i}. \tag{38}$$

Unlike HCA, which requires an exact match along the entire path, POR allows partial correctness by averaging the proportion of correctly predicted nodes. This provides a fine-grained assessment of how well model outputs align with the target hierarchy.

**Strict Point-Overlap Ratio (S-POR).** S-POR refines POR by rewarding only contiguous segments of correct predictions. For the $i$-th sample, we locate the longest run of consecutive correctly predicted nodes and normalize by the hierarchy depth $L_i$:

$$\text{S-POR} = \frac{1}{N} \sum_{i=1}^{N} \frac{1}{L_i} \max_{1 \leq a \leq b \leq L_i} \Big[ (b - a + 1)$$
$$\times \prod_{j=a}^{b} \mathbb{1}\big[ f_\theta(x_i; \mathcal{Y}_j) = y_j^i \big] \Big]. \tag{39}$$

This stricter definition penalizes isolated correct predictions and emphasizes full-path consistency within the hierarchy.

**Top Overlap Ratio (TOR).** Following (Wu et al., 2024), TOR evaluates local hierarchical consistency by considering adjacent layer pairs as independent evaluation units:

$$\text{TOR} = \frac{1}{N} \sum_{i=1}^{N} \frac{1}{L_i - 1} \sum_{j=1}^{L_i - 1} \mathbb{1}\big[ f_\theta(x_i; \mathcal{Y}_j) = y_j^i \big]$$
$$\times \mathbb{1}\big[ f_\theta(x_i; \mathcal{Y}_{j+1}) = y_{j+1}^i \big]. \tag{40}$$

A TOR value of 1 indicates perfect pairwise consistency between consecutive layers, while lower scores reveal local violations of the hierarchical structure.

### B.2. Evaluation Setting

For each setting, we treat large multimodal models (LMMs) as image classifiers $f_\theta$ and leverage language prompts to query predictions at different taxonomy levels. Following (Tan et al., 2026), we formulate HVR as a set of level-specific VQA tasks $(x^i, \mathcal{Y}_j)$, where $i = 1, \ldots, N$ indexes images and $j = 1, \ldots, L^i$ denotes taxonomy depth. To enable closed-set evaluation and avoid the ambiguity of open-set generation (Zhang et al., 2024), each task is cast as a four-choice VQA problem:

> <image>Given the plant in the image, what is its taxonomic
> classification at the <hierarchy>(e.g., kingdom) level?
> A.<similar class>    B.<ground truth>
> C.<similar class>    D.<similar class>
> Answer with the option letter only.

Although four-choice VQA is simpler than conventional hierarchical classification, whose label space grows exponentially with depth, we increase task difficulty by constructing semantically confusing distractors. Specifically, for each taxonomy level, we compute cosine similarity between the image and all incorrect labels using SigLIP (Zhai et al., 2023), and select the top three most similar labels as distractors. This ensures that all options lie at the same hierarchy level and exhibit strong semantic overlap with the ground truth.

### B.3. Implementation Details

All experiments are performed on 4×A6000 GPUs with a batch size of 1 per GPU and a two-step gradient accumulation. The models are all trained for 10 epochs. $\lambda_{\text{ent}}$ and $\lambda_{\text{disp}}$ are set to 0.1 and 0.01 by default. Curvature-aware scaling is applied to image embeddings with curvature $\kappa = 0.05$.

*Table 7.* Loss Coefficient $\lambda_{\text{ent}}$ and $\lambda_{\text{disp}}$.

|  | $\lambda_{\text{ent}} = 0.05$ | $\lambda_{\text{ent}} = 0.1$ | $\lambda_{\text{ent}} = 0.2$ |
|---|---|---|---|
| $\lambda_{\text{dis}} = 0.005$ | 17.40 | 16.22 | 17.18 |
| $\lambda_{\text{dis}} = 0.01$ | 16.55 | 18.07 | 16.77 |
| $\lambda_{\text{dis}} = 0.02$ | 16.45 | 17.55 | 16.25 |

## B.4. Effects of Loss Coefficients $\lambda_{\text{ent}}$ and $\lambda_{\text{disp}}$

We study the impact of the loss weights $\lambda_{\text{ent}}$ and $\lambda_{\text{disp}}$ in Table 5, Overall, all configurations studied improve upon the SFT baseline (HAC on base classes: 14.06), further suggesting that the regularizer is broadly effective. Model performance peaks at $\lambda_{\text{ent}} = 0.1$ and $\lambda_{\text{disp}} = 0.01$. Smaller values weaken the hierarchical and discriminative signals, while larger values can over-constrain the model and interfere with the primary language modeling objective.

