# OpenReview forum: "Learning Taxonomic Trees with Hierarchical Representation Regularization for Large Multimodal Models"
_ICML.cc/2026/Conference — ICML 2026 regular_

### Official Review · Reviewer_6VpQ · 2026-02-25

**Soundness:** 3
**Presentation:** 3
**Significance:** 3
**Originality:** 3
**Overall Recommendation:** 4
**Confidence:** 3

**Summary:**

This work introduces Hierarchical Representation Regularizaton (HiR$^2$), a framework for fine-tuning LMMs on hierarchical visual recognition tasks. Given a hierarchy over a set of fine-grained labels, the objective in these tasks is to predict not simply a single fine-grained class, but a complete ancestral path to this class within the given hierarchy. The proposed framework consists of two key components: 1) the taxonomic entailment loss, which uses a visual tree that is constructed by applying cross-attention between the textual embeddings of each level of the hierarchy and the visual embeddings from the last layer; and 2) the discriminative dispersive loss, which encourages angular separation at the sibling-level. Based on the experiments, this method results in significantly improved hierarchical consistency when added on top of existing fine-tuning methods.

**Compliance With Llm Reviewing Policy:**

Affirmed.

**Final Justification:**

Based on the paper and the rebuttal, I think the method works quite well and is interesting enough for publication. However, as mentioned in my review, I think the applicability of the method is slightly limited, which is why I chose to give a weak accept recommendation.

**Key Questions For Authors:**

1. In light of the aforementioned weakness, would it be possible to provide an ablation for the curvature parameter and for the formulation of the curvature-aware scaling?
2. I do not fully understand how the memory bank works. For a given parent, does it contain embeddings of sibling? How (often) are these embeddings updated?
3. What were all the leaf accuracies in the experiments? I suppose, based on Figure 3, that these were quite low and understandably so, given the difficulty of fine-grained classification on these datasets, but for completeness it would still be interesting to see these in my opinion.

**Limitations:**

Yes.

**Strengths And Weaknesses:**

**Strengths**
- The idea for generating a visual tree through the proposed cross-attention approach is very interesting and seems to work well.
- The paper is well-structured.
- The ablations regarding the dispersive loss are quite extensive.
- The method performs well and is robust across LMMs and taxonomies.

**Weaknesses**
- If my understanding is correct, then the method requires labels at each level of a predefined taxonomy for every training instance. It seems to me that this makes the method slightly difficult to apply in practice in many cases, making its application potential a bit limited.
- The evaluation protocol seems a bit problematic to me, as it can only be used to make predictions when the ground truth is known. It would be nice to see how the method performs under a setting that can be applied to data for which labels are unknown.
- The curvature-aware scaling seems to counteract the exponential growth of hyperbolic space for curvatures smaller than -1. For such curvatures, the embeddings are scaled down a lot before being mapped using the exponential map. This results in reduced average distance from the embeddings to the hyperbolic origin, while the exponential growth of volumes in this space is with respect to a linear increase in hyperbolic radius. This could also explain why the authors found such a low curvature ($-\kappa = -0.05$) to work best. Did the authors experiment with higher curvatures and without the curvature scaling term?

---

> ### Author Rebuttal · Authors · 2026-03-31
>
> Thank you for your supportive review and suggestions. Below we respond to the comments in **Weaknesses (W)** and **Questions (Q)**.
>
> > ***W1: Dependence of labels at each level of a predefined taxonomy.***
>
> We agree that our method assumes a predefined taxonomy and the ancestral path for each training instance, since our target setting is HVR rather than flat classification. However, it can be easily applied in practice for two reasons:
> (1) **Data are commonly organized hierarchically**: Prior study [a] has shown that computer vision datasets are commonly organized hierarchically with broad everyday visual taxonomies, like the WordNet hierarchy behind ImageNet, the biological ontology of birds in CUB, or the tree of verbs in Kinetics.
> (2) **Our method is robust to noisy or partially incorrect taxonomies**: We test our method under taxonomies with three new settings: **Partial-20** (20% incomplete hierarchies), **InternalNoise-10** (10% wrong internal nodes), and **Mixed-20** (20% mixed corruption). Under the same training protocol, **our method remains strong in all three cases**, with only small drops from clean labels and still clear improvements over SFT (P:17.30, I:17.44, M:17.08, SFT:15.74). This indicates our method is robust to moderate incompleteness and taxonomy noise, not dependent on perfectly clean trees.
>
> > ***W2: Open-ended evaluation.***
>
> Thank you for emphasizing the importance of open-ended evaluation. Instead of the four-choice VQA protocol, we prompt the model to generate the full taxonomic path. SFT+Ours again consistently surpasses SFT, demonstrating that our method meaningfully **improves hierarchical understanding in broader LMM usage**. Detailed results on iNat-Animal novel classes are as follows:
>
> | Method  | HCA | POR | S-POR | TOR |
> |---|---|---:|---:|---:|
> | SFT | 12.25 | 70.63 | 69.40 | 64.87 |
> | SFT + Ours | **16.56** | **71.70** | **70.19** | **66.04** |
>
> > ***W3&Q1: Higher curvatures without the curvature scaling term.***
>
> Without the curvature scaling term, as curvature increases (0.05 → 0.1 → 0.2), HCA changes on novel classes (15.71 → 17.39 → 16.40). Our proposed curvature scaling is used to stablize the training process and achieves a better result (w scaling: 17.60, w/o scaling: 15.71 under the same curvature 0.05).
>
>
> > ***Q2: Details of memory bank.***
>
> For each layer and each parent_id, the memory bank stores **past detached image embeddings from samples that share that parent**, so in practice they are embeddings of that parent’s child categories, i.e., potential siblings under the same parent. It is **updated every training step** whenever that parent appears in the current batch: the code appends the current embeddings for that parent and keeps only the most recent entries of a default size 256. During the loss, current embeddings are compared only with memory from the **same parent**, after angular normalization.
>
> > ***Q3: Leaf node accuracy.***
>
> Thanks for emphasizing the importance of including $Acc_{\text{Leaf}}$ for result completeness. We provide Qwen2.5-VL-3B results on iNat-Animal and iNat-Plant base classes in terms of $Acc_{\text{Leaf}}$ here, and will include the complete results in the revised paper.
>
> | Method        | iNat-Animal | iNat-Plant |
> |--------------|-------------:|------------------:|
> | SFT          |   40.37          |   39.95            |
> | SFT + Ours   |   **41.37**          |     **40.70**          |
> | DFT          |   40.52          |       39.72        |
> | DFT + Ours   |   **40.67**          |      **40.80**         |

---

> > ### Author Rebuttal · Reviewer_6VpQ · 2026-04-02
> >
> > I would like to thank the authors for their interesting additional experiments and clarifications, which have resolved most of my concerns. The only remaining concern I have is with the seemingly counterproductive effect of the curvature-aware scaling for larger curvatures. However, my most important concerns have been addressed, so I will maintain my rating.

---

> > > ### Author Response · Authors · 2026-04-07
> > >
> > > We thank the reviewer for the thoughtful follow-up question. We additionally conduct ablation study on curvature-aware scaling when curvature is 0.3, 0.5, and 0.8 (much larger than the default 0.05). Detailed ablation results on iNat-Animal novel classes are as follows, confirming that **curvature-aware scaling remains effective even for larger curvatures**.
> > >
> > > | Method | Curvature  | HCA | POR | S-POR | TOR |
> > > |---|---|---|---:|---:|---:|
> > > | w/o scaling | 0.3 | 16.52 | 72.69 | 62.66 | 59.88 |
> > > | **w scaling** | **0.3** | **17.42** | **73.98** | **65.43** | **62.27** |
> > > | w/o scaling | 0.5 | **16.41** | 72.69 | 62.45 | 59.93 |
> > > | **w scaling** | **0.5** | 16.26 | **72.92** | **63.55** | **60.78** |
> > > | w/o scaling | 0.8 | 15.93 | 72.55 | 62.41 | 59.61 |
> > > | **w scaling** | **0.8** | **17.08** | **73.15** | **63.35** | **60.65** |

---

### Official Review · Reviewer_CuAW · 2026-03-05

**Soundness:** 3
**Presentation:** 3
**Significance:** 2
**Originality:** 2
**Overall Recommendation:** 4
**Confidence:** 3

**Summary:**

In this paper, authors propose Hierarchical Representation Regularization (HiR$^2$), a framework designed to improve hierarchical visual recognition (HVR) in large multimodal models (LMMs). The core idea is to augment standard language modeling fine-tuning with geometry-aware auxiliary losses that encourage internal visual representations to respect hierarchical taxonomies.
The approach dwells on constructing semantic-aware visual trees, which extracts hierarchy-aligned visual features from intermediate LMM layers using textual queues. Regularizer has two core components:

i. Taxonomic entailment loss, implemented in hyperbolic (Lorentz) space to encode parent–child containment via entailment cones, which ensures embedding into a tree like topology,

ii. Spherical angular dispersive loss, which enforces sibling-level discrimination without disturbing radial hierarchy encoding.

These components are combined in the loss function with tunable parameters. Extensive experiments are conducted on iNaturalist-2021 (Plant and Animal taxonomies) and CUB-200 across multiple LMM backbones and fine-tuning strategies (SFT and DFT). Results show consistent improvements in hierarchical consistency metrics such as HCA, POR, S-POR, and TOR.

**Compliance With Llm Reviewing Policy:**

Affirmed.

**Key Questions For Authors:**

* How sensitive is performance to curvature (κ) and regularization weights?
* Can the authors provide representation visualizations to verify coarse-to-fine semantic structure?
* How robust is the approach to noisy or partially incorrect taxonomies?
* How exactly does the parent-aware memory bank work? The paper doesn't give details on its size, how it updates, or if training stability depends heavily on it.
*  Is the curvature-aware scaling stable given the very small batch sizes (e.g., a batch size of 1 per GPU mentioned in the appendix)? Does estimating a batch-wise norm on such a tiny batch introduce noise that hurts training?
* How does HiR$^2$ perform in a true open-ended generation setting? The evaluation relies on a simplified 4-choice VQA task, which is much easier than generating the full taxonomic path from scratch.
* Does the standard language modeling loss interfere with the learned hyperbolic geometry? The proofs show the dispersive loss doesn't disrupt the hierarchy, but it's unclear if the standard next-token prediction gradients mess up those embeddings during training.
* How dependent is the method on the base model's initial text-image alignment? If the base model doesn't already have good representations for a novel category, does the cross-attention still manage to extract useful coarse-to-fine features? What is the computational overhead compared to vanilla fine-tuning?


As a side note, Figure 1 is hard to follow, and significantly overlaps with Fig 2. I suggest either removing Figure 1 entirely and referring the reader to Figure 2 in the introduction, or including a more explanatory yet higher-level figure that explains the paper's overall idea instead of Figure 1. There’s a missing reference in line 380 in the second column of the text.

**Limitations:**

There was no mention of limitations of the method, and the impact statement was a boilerplate text. The paper would benefit from stating the limitations and potential expansions, such as the assumption of clear taxonomic structure in the dataset and lack of an analysis of noisy data with incomplete hierarchies etc.

**Strengths And Weaknesses:**

**Strengths:**

The paper takes on a well-known blind spot in large multimodal models: they struggle to accurately grasp hierarchical visual recognition (HVR). The authors rightly point out that standard language training just doesn't teach these models how to understand real-world taxonomies. To fix this, their idea to mix two different geometric spaces is brilliant. First, they use a hyperbolic space (the Lorentz model) to group parent and child categories together, which works perfectly because this kind of space grows exponentially. Second, they use a spherical space to push sibling categories apart—so the model can easily tell them apart—without messing up the parent-child hierarchy they just built.

The resulting HiR$^2$ framework is practically plug-and-play. By pulling visual features from the layers of the LLM using a non-parametric cross-attention trick, they manage to inject taxonomic knowledge into the model without adding any new parameters to train. The math behind the approach is rock solid, with Theorems 4.1 through 4.3 proving that their spherical separation trick won't accidentally break the core hierarchy. On top of that, their testing is super thorough—they proved the method works across a bunch of different model architectures (like Qwen2.5-VL, Intern3.5-VL-1B, and LLaVA-OV) and training styles (both SFT and DFT).

**Weakness**

While the paper’s core idea is technically solid and empirically supported, there are a few areas that need more polish and justification.

- The authors claim their mechanism extracts "coarse-to-fine semantics" from the intermediate layers, but they don't actually provide empirical analysis or visual representations to prove that lower-level features correspond to coarser categories and deeper layers to finer ones. Also, calling this the "first principled study" to introduce hierarchical representation learning to LMMs feels like a bit of an overstatement; they should probably tone that language down or add a solid literature comparison to back it up.

- Another issue is their claim that the method is an "additional lightweight regularization term" that incurs "little computational overhead". They never actually evaluate or report the practical memory and time costs compared to standard fine-tuning to prove this point.

- On the technical side, while they do ablate some loss weights in the appendix, the sensitivity to the hyperbolic curvature parameter $\kappa$ (which is just fixed to 0.05) is totally underexplored. Furthermore, their math is great for showing radial gradient behavior, but the theoretical analysis doesn't touch on the overall convergence properties of putting all these different loss objectives together.

- There are a few experimental design choices that hold the paper back. To evaluate hierarchical visual recognition, they simplify the task into a multiple-choice VQA problem with only four options. Even though they pick tricky, semantically similar distractors, this is way easier than open-ended generation or standard hierarchical classification where the choices grow exponentially. It's also worth noting they mainly tested on nature-focused datasets like iNaturalist and CUB-200—testing on broader, everyday visual taxonomies would make the claims much stronger. To top it off, they completely skipped writing a real limitations section, instead just dropping in a boilerplate impact statement saying there are no societal consequences to highlight.

- The non-parametric cross-attention mechanism used to pull visual features assumes that the visual tokens are already somewhat aligned with the textual queries before the regularization even kicks in. It would be helpful to see how this handles situations where the initial visual-text alignment is poor. Plus, the dispersive loss relies on a "parent-aware memory bank" to keep things stable and diverse. However, there is no real discussion or ablation on how the size or update strategy of this memory bank impacts training. Does it cause instability in the early epochs? We just don't know.

- The whole framework heavily relies on having a perfectly structured, rigid text hierarchy provided during fine-tuning. Real-world data is messy, and taxonomies can be incomplete or overlapping. The authors don't explore how robust HiR$^2$ is to noisy or partially incorrect taxonomic trees, which is a pretty significant practical limitation for a method designed to map the real world.

---

> ### Author Rebuttal · Authors · 2026-03-31
>
> Thank you for the supportive review. Below we address the comments in Weaknesses (W) and Questions (Q).
>
> > W1-1&Q2: Coarse-to-fine semantics.
>
> We provide visualizations in https://anonymous.4open.science/r/Coarse-to-fine-Representation-Visualization-B122/. They show coarse-level clustering at higher taxonomic levels and finer separation at deeper levels, supporting our claim that the learned features follow the taxonomy’s coarse-to-fine structure.
>
> > W1-2: Literature comparison.
>
> We position our method by carefully comparing with the prior work.
>
> 1. Hierarchy-aware representation learning in CLIPs: Prior hierarchy-aware methods such as ProTeCt/HGCLIP/HiCLIP mainly operate in CLIP-like contrastive VLMs, whereas our method targets taxonomy injection in a generative LMM.
>
> 2. Hierarchy-agnostic Representation Learning in LMMs: Compared to VIRAL and JARVIS that add extra hierarchy-agnostic visual regularization, our method instead introduces taxonomy-specific design.
>
> > W2&Q8-2: Training overhead.
>
> Our method adds negligible overhead: throughput is unchanged (0.111 vs. 0.111 steps/s), runtime is nearly identical (4224.31s vs. 4235.86s), and peak GPU memory increases by only 0.002 GiB (16.1152 vs. 16.1172 GiB, about 2 MB). This supports our claim that the added regularization is lightweight.
>
> > W3-1&Q1-1: Curvature ablation.
>
> Without curvature-aware scaling, increasing curvature (0.05 → 0.1 → 0.2) achieves no significant gains on novel-class HCA (15.71 → 16.03 → 15.29) compared to SFT (15.74). With scaling, training is more stable and performance is better overall under the default curvature 0.05 (17.60 vs. 15.71).
>
> > W3-2: Convergence properties.
>
> In the revision, we will add a standard nonconvex result showing that, under smoothness and bounded-variance assumptions, SGD converges to a stationary point for the combined loss. Our current analysis complements this by showing that $L_{\mathrm{disp}}$ does not interfere with the radial hierarchy enforced by $L_{\mathrm{ent}}$, which helps explain stable joint optimization in practice.
>
> > W4-1&Q6: Open-ended evaluation.
>
> We now evaluate a true open-ended setting where the model generates the full taxonomic path without candidates. SFT+Ours again surpasses SFT on iNat-Animal novel classes (HCA: 12.25 → 16.56, POR: 70.63 → 71.70, S-POR: 69.40 → 70.19, TOR: 64.87 → 66.04).
>
> > W4-2: More taxonomies.
>
> We additionally evaluate on Food101 beyond nature-focused taxonomies. SFT+Ours again yields consistent gains on novel classes (HCA: 40.48 → 52.38, POR: 72.62 → 82.54, S-POR: 50.20 → 61.71, TOR: 53.17 → 67.46), suggesting that the method can be applied to broader, everyday visual taxonomies.
>
> > W5-1&Q8-1: Dependence on initial text-image alignment.
>
> Prior analysis [a] shows that LMMs integrate visual information progressively: lower layers capture global signals, while deeper layers inject more question-relevant cues, enabling stronger text-image alignment. As shown in Table 3 in the manuscript, our method is effective across multiple base models, suggesting robustness to differences in initial alignment quality.
>
> > W5-2&Q4: Memory bank details.
>
> The parent-aware memory bank is a per-layer, per-parent FIFO queue for the sibling-dispersive loss. It stores recent detached embeddings and compares only samples under the same parent. In practice, early instability is unlikely because stored entries receive no gradients and the loss is gradually activated. Its main effect is better fine-grained discrimination: increasing the bank size from 16 to 64 raises $Acc_{\text{leaf}}$ from 41.33 to 41.85.
>
> > W6&Q3: Noisy or partially incorrect taxonomies.
>
> We evaluate our method under three corrupted-taxonomy settings: Partial-20 (20% incomplete hierarchies), InternalNoise-10 (10% incorrect internal nodes), and Mixed-20 (20% mixed corruption). Our method only drops slightly with incomplete hierarchies, and still achieves clear gains over SFT (P:17.19, I:18.74, M:18.02, SFT:15.74), indicating robustness to moderate incompleteness and taxonomy noise.
>
> > Q1-2: Regularization weights.
>
> The ablation on $\lambda_{\mathrm{ent}}$ and $\lambda_{\mathrm{disp}}$ is provided in Appendix B.4.
>
> > Q5: Stability of curvature-aware scaling.
>
> Even with batch size 1 per GPU, the scaling factor is computed from multiple hierarchy-level embeddings within each sample rather than a single vector. The average norm is detached and used only as a shared rescaling factor, so its noise is not backpropagated.
>
> > Q7: ‌Interference‌ with $L_{\mathrm{LM}}$.
>
> Since $L_{\mathrm{ent}}$ and $L_{\mathrm{disp}}$ are regularization terms, $L_{\mathrm{LM}}$ is not geometry-preserving in general. However, the consistent performance gains suggest these regularizers are sufficient to counterbalance unconstrained LM gradients in practice.
>
> > W4-3&Q9: Limitations.
>
> We will revise Figure 1 and add a detailed Limitations section.
>
> [a] Cross-modal Information Flow in Multimodal Large Language Models, CVPR 2025.

---

> > ### Author Rebuttal · Reviewer_CuAW · 2026-04-01
> >
> > While the rebuttal provides welcome empirical snapshots, it does not fundamentally bridge the gap between the proposed geometric theory and the practical behavior of the model during training. The following concerns /questions remain critical:
> >
> > - Are there mechanistic ambiguity in (W1-1, Q2)? The authors provided external visualizations of "coarse-to-fine" clustering, but these represent a static "after-the-fact" snapshot. A short rebuttal cannot prove that the model consistently utilizes these hierarchies during the actual reasoning process across diverse prompts. There remains a risk that the model is simply "memorizing" the geometry for the specific taxonomic labels provided rather than developing a generalized hierarchical understanding.
> >
> > - Is there is a conflict in geometric argument  (Q7)? The authors acknowledge that the standard Language Modeling (LM) loss is not geometry-preserving. While they claim $HiR^{2}$ "counterbalances" this, there is no formal analysis of the gradient competition between the next-token prediction objective and the hyperbolic/spherical constraints. Without understanding how these opposing gradients interact, the stability and scalability of $HiR^{2}$ to much larger models or more complex tasks remain speculative.
> >
> > - Is there a simplification of visual attention (W5-1, Q8-1)? The non-parametric cross-attention assumes that visual tokens are already semantically aligned enough with textual queries to form a tree. This bypasses the harder problem of learning the alignment when the model has no prior knowledge of a category, potentially limiting the framework's utility in truly "novel" or zero-shot scenarios where textual cues are insufficient.
> >
> > In conclusion, addressing these points would require a significant shift in the paper's scope—moving from regularization that happens to work to a principled study of geometric gradient interaction. Also, while not critical, there is a dependency on rigid structure's. The tests on "noisy" taxonomies are a good start, but they still assume a predefined, tree-like structure. In the real world, visual concepts often overlap or form Directed Acyclic Graphs (DAGs) rather than strict trees. The current framework's reliance on a fixed hyperbolic Lorentz model is mathematically tied to tree structures, and its performance on fluid or non-hierarchical "common sense" relations has not been established. Therefore, these core concerns are not easily resolved within the constraints of the rebuttal period.

---

> > > ### Author Response · Authors · 2026-04-07
> > >
> > > We thank the reviewer for the thoughtful follow-up questions. Below, we provide additional evidence on mechanism, gradient interaction, and alignment robustness, and clarify the current scope of the method.
> > >
> > >
> > > > ***Q1. Inference-time use of hierarchy.***
> > >
> > > We agree that post-hoc visualization alone cannot show whether the model actually uses hierarchy during inference. To address this, we add an inference-time causal intervention on $HiR^2$ and the SFT baseline over 24 samples of iNat-Animal novel classes and 4 prompt templates (96 cases per source-target pair). For each hierarchy level, we ablate the corresponding level-conditioned image tokens at different decoder layers during teacher-forced inference and measure token-level NLL (negative log-likelihood) for each predicted taxonomy level. We summarize selectivity by
> > > $$
> > > \Delta NLL_{\mathrm{diag}}-\Delta NLL_{\mathrm{off}},
> > > $$
> > > where a positive value means stronger same-level disruption. Layer 7 mainly causes broad degradation in both models, consistent with removing generic visual evidence. **At intermediate layers, $HiR^2$ shows stronger level selectivity than SFT**: layer 14, $+0.017$ vs. $+0.013$; layer 21, $+0.012$ vs. $+0.008$. The effect becomes weak again at layer 28. While this is not a full proof of hierarchical reasoning, it provides causal evidence that $HiR^2$ uses hierarchy-aware representations during inference rather than only exhibiting a post-hoc geometry.
> > >
> > > > ***Q2. LM-vs-geometry gradient interaction.***
> > >
> > > We agree that the LM objective is not geometry-preserving in isolation. We therefore diagnose gradients at different checkpoints (early: 100, mid:300, late: 470) on the image-token hidden states of the top two layers. On the penultimate layer, **the LM and geometric gradients remain nearly orthogonal throughout training**:
> > > $$
> > > \cos(g_{LM},g_{ent})=0.00110/0.00009/-0.00120,
> > > $$
> > > $$
> > > \cos(g_{LM},g_{disp})=-0.00064/0.00002/-0.00123.
> > > $$
> > > At the final regularized layer, the LM gradient on the measured image-token states is effectively zero, while **the geometric gradients remain active**:
> > > $$
> > > \|g_{LM}\|=0,
> > > $$
> > > $$
> > > \|g_{ent}\|=4.90\times10^{-4}/8.43\times10^{-4}/1.26\times10^{-3}.
> > > $$
> > > **These results suggest complementarity rather than destructive competition** between the LM and geometric objectives at the current scale. We agree that a formal theory for larger models remains beyond the scope of the rebuttal, and we will state this limitation explicitly.
> > >
> > > > ***Q3. Robustness to imperfect alignment.***
> > >
> > > We agree that $HiR^2$ is not designed to learn vision-language alignment from scratch. We therefore test two harder settings:
> > >
> > > (1) Query-only misalignment during training: We replace 10% or 20% of the hierarchy queries used by $HiR^2$ with sibling-category queries while keeping SFT supervision unchanged. Table 1 shows that performance gain compared to SFT remains highly stable, suggesting **robustness to moderate local semantic misalignment where visual tokens are not semantically aligned enough with textual queries**.
> > >
> > > (2) Cross-domain transfer: We train on iNat-Animal base classes and evaluate on iNat-Plant novel classes, where they do not share taxonomies. Table 2 shows that $HiR^2$ still consistently outperforms SFT under both taxonomy and visual-distribution shift. This supports **usefulness in truly "novel" or zero-shot scenarios**.
> > >
> > > **Table 1. Query-only misalignment ablation on animal novel.**
> > >
> > > | Model | HCA | POR | S-POR | TOR |
> > > |---|---:|---:|---:|---:|
> > > | SFT | 15.74 | 72.12 | 61.42 | 58.86 |
> > > | Ours (10% misaligned queries) | 18.05 | **74.15** | **65.42** | **62.31** |
> > > | Ours (20% misaligned queries) | **18.16** | 73.68 | 64.27 | 61.61 |
> > >
> > > **Table 2. Cross-domain transfer.**
> > >
> > > | Model | HCA | POR | S-POR | TOR |
> > > |---|---:|---:|---:|---:|
> > > | SFT | 13.44 | 66.54 | 50.68 | 49.21 |
> > > | Ours | **13.82** | **68.11** | **52.32** | **51.24** |
> > >
> > > > ***Q4. Dependence on tree-structured taxonomies.***
> > >
> > > We thank the reviewer for this important point. Our paper **focuses on hierarchical visual recognition (HVR), where the main practical setting is a tree-structured taxonomy**, as in biologically grounded benchmarks such as iNat and CUB, which follow coarse-to-fine category chains (e.g., kingdom → phylum → class → order → family → genus → species).
> > >
> > > We agree that DAG-structured ontologies exist, for example in WordNet/ImageNet-style semantic ontologies or cross-listed product catalogs. However, they are not the primary target of this work. In principle, the cone-based entailment formulation can be extended by enforcing all valid parent-child edges, but we do not claim a systematic study of multi-parent DAG supervision or of non-hierarchical commonsense relations here. **We will make this scope boundary explicit in the revision: $HiR^2$ is designed for tree-like taxonomic hierarchies, and we will extend to DAGs or more general relational knowledge in future work.**

---

### Official Review · Reviewer_DXzd · 2026-03-13

**Soundness:** 2
**Presentation:** 3
**Significance:** 3
**Originality:** 3
**Overall Recommendation:** 4
**Confidence:** 4

**Summary:**

This paper studies hierarchical visual recognition in large multimodal models, focusing on the common failure mode where a model may produce a plausible fine-grained prediction while remaining inconsistent with the underlying taxonomy. To address this, the paper proposes HiR2, a lightweight regularization framework added during multimodal fine-tuning.

**Compliance With Llm Reviewing Policy:**

Affirmed.

**Final Justification:**

The additional clarifications and experimental results has fully addressed my concerns. Thus I keep my score as "weak accept".

**Key Questions For Authors:**

How does HiR2 compare against stronger hierarchy-aware baselines or simpler alternatives? In particular, I would like to see comparisons against path-level prompting, level-wise auxiliary supervision, or prior taxonomy-aware VLM/LMM methods. A convincing comparison here would materially improve my assessment of both originality and significance.

Do the gains persist in a more realistic evaluation setting than four-choice VQA?

Does the method preserve general multimodal ability outside taxonomy benchmarks? Since the paper emphasizes a plug-and-play regularizer with minimal interference, I would like to see at least a small no-regression check on standard multimodal tasks. Positive evidence here would strengthen the practical case a lot.

How much of the novel-class gain depends on shared ancestor structure between base and novel leaves? A stricter split by held-out branches or subtrees would clarify whether the method truly helps hierarchical generalization rather than mainly exploiting shared upper-level taxonomy. Strong results under such a split would improve my evaluation.

What is the actual training and memory overhead of the added regularization? Since efficiency and ease of integration are part of the selling point, even a small overhead table would make the practical contribution much more convincing.

**Limitations:**

No.

The paper includes only a very generic discussion of limitations and societal impact

**Strengths And Weaknesses:**

Strengths

Relevant problem. Hierarchical consistency is a genuine weakness of current multimodal LLMs, especially in fine-grained recognition settings. The paper addresses a problem that is important and timely.

Clear high-level intuition. The core idea is easy to understand: encourage the model’s internal visual representations to reflect taxonomic structure rather than relying only on the language modeling objective. This makes the paper more compelling than work that only adds a task-specific decoding trick.

Weaknesses

Most of the evidence comes from biological taxonomies and a four-choice VQA protocol with curated distractors. This is a controlled benchmark, but it is still quite different from open-ended hierarchical recognition or free-form multimodal generation. Because of this, I am not yet convinced that the method meaningfully improves hierarchical understanding in broader LMM usage.

The paper mainly compares against plain SFT/DFT plus internal ablations. Given the framing, I would have expected stronger comparisons or at least a more careful discussion around hierarchy-aware alternatives such as ProTeCt-style prompt calibration, HGCLIP/HiCLIP-type hierarchy-aware objectives, hyperbolic multimodal representation methods such as MERU and later hyperbolic VLM/LMM extensions, and recent MLLM representation-regularization approaches such as VIRAL or JARVIS. Without such comparisons, it is hard to tell how much of the gain comes from the proposed taxonomy-specific design versus the more general benefit of adding extra visual regularization.

Although the problem is interesting, the current study is still largely confined to natural-world taxonomies with clean hierarchical labels. The paper does not yet show whether the idea transfers to messier, less curated taxonomies or to broader multimodal tasks.

In the current setup, base and novel leaves still share higher-level taxonomy. That is a reasonable protocol, but it makes the generalization story less strong than it may first appear. I would like to see evidence under stricter splits, for example unseen branches or held-out subtrees.

---

> ### Author Rebuttal · Authors · 2026-03-31
>
> Thank you for your supportive review and suggestions. Below we respond to the comments in **Weaknesses (W)** and **Questions (Q)**.
>
> > ***W1&Q2: Open-ended evaluation & More taxonomies.***
>
> Thank you for emphasizing the importance of open-ended evaluation and experiments beyond biological taxonomies. We newly add two sets of experiments:
>
> (1) Open-ended evaluation: Instead of the four-choice VQA protocol, we prompt the model to generate the full taxonomic path. SFT+Ours again consistently surpasses SFT, demonstrating that our method meaningfully **improves hierarchical understanding in broader LMM usage**. Detailed results on iNat-Animal novel classes are as follows:
>
> | Method  | HCA | POR | S-POR | TOR |
> |---|---|---:|---:|---:|
> | SFT | 12.25 | 70.63 | 69.40 | 64.87 |
> | SFT + Ours | **16.56** | **71.70** | **70.19** | **66.04** |
>
> (2) More taxonomies: We additionally evaluate on Food101 beyond nature-focused taxonomies. Our method again yields consistent gains on novel classes, suggesting that the method **can be applied to broader, everyday visual taxonomies**. Detailed results on Food101 novel classes are as follows:
>
> | Method  | HCA | POR | S-POR | TOR |
> |---|---|---:|---:|---:|
> | SFT | 40.48 | 72.62 | 50.20 | 53.17 |
> | SFT + Ours | **52.38** | **82.54** | **61.71** | **67.46** |
>
> > ***W2&Q1: Comparison against stronger baselines.***
>
> Thank you for pointing out these important related directions.
>
> 1. Hierarchy-aware alternatives: Prior hierarchy-aware methods such as ProTeCt/HGCLIP/HiCLIP mainly operate **in CLIP-like contrastive VLMs**, whereas our method targets taxonomy injection in **a generative LMM**. We further compare with simple hierarchy-aware alternatives, including path-level prompting and level-wise supervision, and show that **hierarchical labels alone are not sufficient**, while our method brings additional gains by explicitly regularizing the representation structure.
>
> 2. Hyperbolic LLMs/LMMs: Recent Hyperbolic LLM/LMM work [a,b] mainly studies hyperbolic representations for LLMs/LMMs while preserving stable training. By contrast, our method uses hyperbolic geometry only during training, while **keeping the original autoregressive decoding pipeline in the standard Euclidean hidden space unchanged** at inference time, making it compatible with mainstream LMMs.
>
> 3. Generic visual regularizers: Comparisons with VIRAL and JARVIS show that our gains are **not simply due to extra visual supervision**, but come from injecting taxonomy-specific structural priors into LMM hidden states.
>
> | Method | HCA | POR | S-POR | TOR |
> |---|---:|---:|---:|---:|
> | Path-level Prompting | 3.97 | 41.86 | 35.24 | 31.72 |
> | Level-wise Supervision | 16.84 | 73.15 | 64.13 | 60.63 |
> | SFT + VIRAL | 11.40 | 67.95 | 52.46 | 53.17 |
> | SFT + JARVIS | 12.96 | 70.79 | 58.91 | 56.70 |
> | SFT + Ours | **17.60** | **73.77** | **64.63** | **61.49** |
>
> > ***W3: Experiements on messier, less curated taxonomies.***
>
> We understand the reviewer's concern about whether the idea transfers to messier, less curated taxonomies. We test HiR under noisy taxonomies with three new settings: **Partial-20** (20% incomplete hierarchies), **InternalNoise-10** (10% wrong internal nodes), and **Mixed-20** (20% mixed corruption). Under the same training protocol, **our method only drops slightly with incomplete hierarchies, and still achieves clear gains over SFT** (P:17.19, I:18.74, M:18.02, SFT:15.74). This indicates our method is robust to moderate incompleteness and taxonomy noise, not dependent on perfectly clean trees.
>
> > ***W4&Q4: Stricter generalization.***
>
> We understand the reviewer's concern about the generalization since the base and novel levels still share higher-level taxonomy. We newly train on iNat-Animal base classes and evaluate on iNat-Plant novel classes. The consistent performance gains (HCA: 13.44 → 13.82, POR: 66.54 → 68.11, S-POR: 50.68 → 52.32, TOR: 49.21 → 51.24) show that the method **truly helps hierarchical generalization rather than mainly exploiting shared upper-level taxonomy**.
>
> > ***Q3: General multimodal ability.***
>
> To comprehensively assess the model’s general capabilities endowed with FGVR capability, we newly evaluate on a classification-based VQA benchmark: ImageWikiQA. Results (SFT: 53.10%, SFT+Ours: 54.00%) demonstrate that our method has strong classification ability **while preserving general multimodal question-answering ability**.
>
> > ***Q5: Training and memory overhead.***
>
> Thank you for emphasizing the importance of practical efficiency. Our method adds negligible overhead: throughput is unchanged (0.111 vs. 0.111 steps/s), runtime is nearly identical (4224.31s vs. 4235.86s), and peak GPU memory increases by only 0.002 GiB (16.1152 vs. 16.1172 GiB, about 2 MB). This supports our claim that **the added regularization is lightweight**.
>
> [a] Hyperbolic Learning with Multimodal Large Language Models, ECCV 2024 Workshop.
>
> [b] Hyperbolic Large Language Models, Arxiv 2025.

---

> > ### Author Rebuttal · Reviewer_DXzd · 2026-04-06
> >
> > Thank you to the authors for the additional clarifications.

---

> > > ### Author Response · Authors · 2026-04-07
> > >
> > > Thank you for reading our rebuttal. We will carefully incorporate all the additional clarifications into the final version to further improve the quality of the paper.

---

### Decision · Program_Chairs · 2026-04-30

**Decision:**

Accept (regular)

**Comment:**

This paper investigates an explicit regularization to make LLMs taxonomically aware. The reviewers are clear and in agreement on the strengths of the paper: it is a clear blind spot of LLMs, the solution is plug-and-play, the paper is well written, and the solution is interesting. In the initial reviewers, several issues were highlighted. The AC found the comments on the evaluation setting (specifically the limited VQA scope), need for stronger baselines, and open world generalization most important. The rebuttal clearly provides an answer to all parts, with new open world protocols and more comparisons. As such, all reviewers recommend acceptance and the AC agrees.